# SILENCE-THE-MIMIC: ACCELERATING IMPERCEPTIBLE PERTURBATIONS AGAINST VOICE CLONING

## ABSTRACT

Deep neural network–based Voice Conversion (VC) and Text-to-Speech (TTS) models have rapidly advanced, enabling realistic voice cloning with minimal input data. Such capabilities raise serious concerns over unauthorized cloning of speaker identities and the associated privacy and security risks. Current imperceptible adversarial protection methods rely on quality control losses that are highly sensitive to hyperparameter tuning and computationally expensive due to lengthy optimization. To address these limitations, we propose a fast yet imperceptible protection method that injects perturbations in the frequency domain under a psychoacoustic masking–based constraint. Our approach strictly enforces perceptibility bounds during adversarial training, eliminating the need for iterative quality balancing and significantly reducing computational cost. Experimental results on multiple state-of-the-art VC and TTS models show that our method achieves protection performance comparable to or better than existing baselines, with at least an order-of-magnitude speedup. These results demonstrate the effectiveness of frequency-domain perturbations with perceptual constraints as a practical paradigm for protecting against voice cloning.

## 1 INTRODUCTION

With the rapid advancement of Deep Neural Networks (DNN), voice cloning models such as Voice Conversion (VC) and Text-to-Speech (TTS) have reached an unprecedented stage of development. Recent VC and TTS models require only a few seconds of a target speaker's audio to generate synthetic speech with high fidelity, closely resembling the speaker's vocal characteristics while allowing arbitrary linguistic content (Bargum et al., 2024; Deng et al., 2025). These powerful models have enabled a wide range of applications, including voice restoration in the film industry, voice assistants in smartphones and Internet-of-Things devices, and accessibility aids for individuals with speech-related disabilities. However, highly realistic synthesized speech, commonly referred to as DeepFake audio, poses serious threats to individuals' privacy, property, and reputation.

In response to these threats, existing research has explored methods to defend against unauthorized DeepFake audio (Wu et al., 2025; Nguyen-Le et al., 2025). These approaches can be broadly categorized into two groups: *detection* and *protection*. DeepFake audio detection focuses on distinguishing machine-generated speech from human speech. This can be achieved by examining artifacts present in synthesized audio (passive detection) (Sun et al., 2023) or by extracting watermarks embedded in the audio beforehand (active detection)(Roman et al., 2024). However, detection-based strategies are effective only after an attack has occurred, which limits their protective capacity. In contrast, protection methods aim to disrupt the speaker identity extraction process, typically carried out by a speaker encoder, in voice cloning models. This is achieved by introducing imperceptible perturbations, such as Gaussian noise or perturbations generated through adversarial training, into the audio before publication Gao et al. (2025). By doing so, these methods prevent the replication of the reference audio's timbre and safeguard the speaker's identity. Such source level protection reduces the risk of voice cloning in advance.

This paper proposes a new methodology that is significantly faster than existing imperceptible white-box adversarial perturbation based protection methods. Most adversarial perturbation based defense algorithms incorporate additional audio quality control losses, which are typically based on measures such as the L2 norm, Signal to Noise Ratio (SNR), spectrogram or Mel spectrogram distances,

equal loudness contours, and psychoacoustic models. Although these methods can inject effective adversarial noise while maintaining audio quality to some extent, they face two major challenges: (i) the outcome of audio quality control is highly sensitive to the relative weighting of the quality loss in the objective function, meaning that when applying these methods to voice cloning models, one must carefully balance the weight between the speaker encoder loss and the quality control loss to achieve satisfactory performance; and (ii) they typically require hundreds or even thousands of iterations for the model to identify an effective perturbed subspace that simultaneously preserves acceptable sound quality (Yu et al., 2023; Li et al., 2023a). Such procedures are computationally expensive and limit their practical applicability.

Our proposed method addresses these limitations by introducing imperceptible perturbation in the frequency domain. Specifically, we first convert the speech signal into a spectrogram in the frequency domain using the Short-Time Fourier Transform (STFT), under the Constant Overlap Add (COLA) condition to ensure lossless inverse STFT (iSTFT) reconstruction. A psychoacoustic model is then applied to the spectrogram to estimate the masking threshold of each tone in every STFT frame. This threshold is combined with a predefined tolerance to establish a strictly enforced human perceptibility constraint, ensuring that the injected perturbations remain within the allowable range. In this manner, gradient updates are directed solely toward the perturbations, which accelerates adversarial training while preserving strong protection performance.

We conduct extensive experiments on multiple state-of-the-art VC and TTS models and compare our approach with leading white-box defense baselines. The results demonstrate that the proposed method achieves comparable or superior protection performance while being at least an order of magnitude faster than existing baselines.

## 2 RELATED WORK

**Voice Cloning Models**  Voice cloning refers to generating speech with arbitrary linguistic content that mimics the vocal characteristics of a specific individual. Recent approaches mainly fall into two paradigms: VC (Qian et al., 2019; Chou et al., 2019; Wang et al., 2021; Li et al., 2021; 2023b; Guo et al., 2023; Park et al., 2023) and TTS (Casanova et al., 2024; Liao et al., 2024; Du et al., 2024; Guo et al., 2024; Chen et al., 2024; RVC-Boss, 2025; Deng et al., 2025). VC models take utterances from two speakers: one provides linguistic content (what to say), and the other provides speaker identity (how to say it). The model then combines these inputs to synthesize speech that carries the content of the former while preserving the vocal traits of the latter. TTS models, on the other hand, generate acoustic features from input text conditioned on a short reference audio that provides speaker information. These features are then passed to a vocoder to synthesize speech that preserves the vocal characteristics of the reference speaker.

**Detection-based Protection**  To mitigate the risks posed by unauthorized voice cloning, researchers have developed proactive algorithms that embed stealth identifiers into audio files to enable post-cloning tracking, a technique commonly referred to as audio watermarking (Chen et al., 2023; Liu et al., 2023; Roman et al., 2024; Liu et al., 2024). Audio watermarking methods typically involve a generator–detector framework: the generator embeds imperceptible patterns into speech, which tend to remain intact even after processing or cloning by VC or TTS models, while the detector recovers or verifies these patterns to trace the audio's origin and assess its authenticity.

Another approach to detecting DeepFake audio is to analyze intrinsic artifacts in model synthesized speech (Sun et al., 2023). These passive methods identify acoustic, spectral, or temporal inconsistencies introduced during synthesis and use statistical models or deep neural networks to distinguish synthetic speech from genuine recordings.

**Adversarial Perturbation Protection**  Another promising defense strategy is to embed imperceptible or minimally perceptible noise into speech audio (Huang et al., 2021; Yu et al., 2023; Li et al., 2023a; Wang et al., 2023; Dong et al., 2024; Yang et al., 2024). Such noise is specifically designed to disrupt the cloning process, particularly the speaker encoder, thereby preventing voice cloning models from synthesizing accurate outputs. Adversarial perturbation based defenses can be categorized along two binary dimensions: how the perturbations are generated and where they are applied. With respect to generation, some methods produce perturbations through adversarial training, while others employ a pretrained perturbation generator that does not require gradient information dur-

ing inference. With respect to application, perturbations are either injected directly into the audio waveform or introduced by biasing latent representations within a vocoder.

## 3 METHODOLOGY

In this section, we present the proposed protection algorithm, Silence-the-Mimic (STM). Unlike prior approaches that rely on perceptual loss functions to regulate the audibility of perturbations, STM employs a psychoacoustic-based soft clamping strategy to enforce strict control over perceptual distortion in the frequency domain. This design enables direct control over the perceptual quality of protected speech while simultaneously accelerating the perturbation process by eliminating the need for additional audio refinement steps.

We begin this section by formulating the protection problem, followed by a detailed description of our methodology.

### 3.1 PROBLEM SETUP

Our problem scenario involves three entities: (i) a victim who publishes publicly available content containing their speech samples, denoted by the set $\mathbb{X}_u$, (ii) an attacker who seeks to exploit these samples to generate deepfake audio, and (iii) a verifier that is exclusively authorized to interact with the victim.

The attacker is modeled as a third-party adversary without the capability to directly capture the victim's speech. Instead, the attacker collects publicly available speech samples of the victim. To generate spoofed speech, the attacker employs a speech synthesis model $G(enc_{\mathrm{spk}}(\boldsymbol{x}), c) : \mathbb{R}^{d_{enc}} \times \mathcal{C} \to \mathbb{R}^*$, which synthesizes an utterance containing linguistic content $c$ while mimicking the vocal characteristics of the target utterance $\boldsymbol{x} \in \mathbb{X}_u$. Here, $\boldsymbol{x} \in \mathbb{R}^*$ denotes a variable-length sequence representing the audio waveform, where the symbol $*$ indicates that the sequence length may vary. $c \in \mathcal{C}$ denotes arbitrary content selected by an attacker to facilitate misuse, and $enc_{\mathrm{spk}} : \mathbb{R}^* \to \mathbb{R}^{d_{enc}}$ denotes the speaker encoder.

The verifier relies on a speaker verification (SV) model $SV : \mathbb{R}^* \to [0, 1]$ to assess whether a given utterance $\boldsymbol{x}$ originates from the victim. Based on the confidence score produced by the SV model, the verifier accepts or rejects an utterance as authentic, thereby serving as the ultimate gatekeeper in controlling access to victim-authorized interactions.

To safeguard the victim's voice identity from replication by an attacker, a protection algorithm perturbs every $\boldsymbol{x} \in \mathbb{X}_u$ prior to publication, thereby altering the audio characteristics such that the SV model no longer recognizes them as identical to the original. At the same time, the perturbation is designed to remain as imperceptible as possible, ensuring that the protected utterance remains intelligible and usable. This process can be formulated as:

$$
\begin{aligned}
\min_{\boldsymbol{\Delta_x}} \ &SV\big(G\big(enc_{\mathrm{spk}}(\boldsymbol{x} + \boldsymbol{\Delta_x}), c\big)\big) \\
\text{s.t. } &H(\boldsymbol{x} + \boldsymbol{\Delta_x}, \boldsymbol{x}) \leq \mathbf{H}_{\mathrm{limit}},
\end{aligned}
\tag{1}
$$

where $H : \mathbb{R}^* \times \mathbb{R}^* \to \mathbb{R}^d$ denotes a human perceptual function that quantifies the dissimilarity between two inputs, with the output dimensionality $d$ may correspond to a scalar or a high dimensional tensor depending on the granularity of perceptual constraints (e.g., time–frequency bins). The parameter $\mathbf{H}_{\mathrm{limit}}$ specifies the maximum allowable distortion and enables fine-grained control across different feature channels. In practice, directly optimizing Equation 1 is often infeasible. Instead, we modify the speaker identity perceived by the synthesizer to achieve a similar outcome, as discussed in a later section.

### 3.2 METHOD FORMULATION

Our proposed method, STM, begins by converting the victim's speech audio into a frequency-domain representation, i.e., a spectrogram. A psychoacoustic model is then applied to constrain the maximum allowable distortion of the protected audio. Finally, the optimization problem in

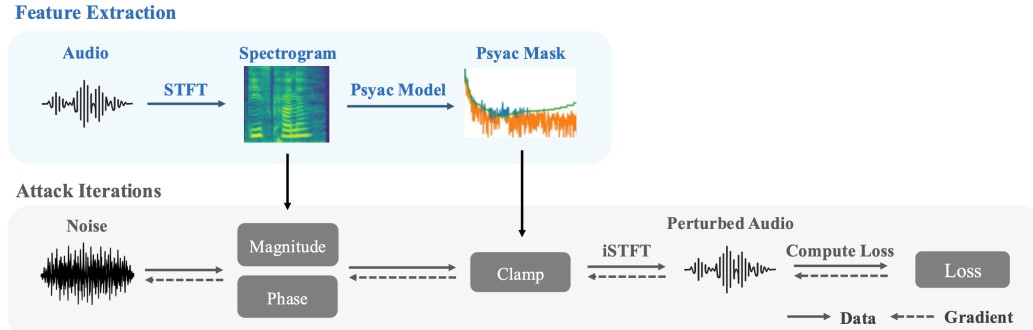

Figure 1: Pipeline of STM.

Equation 1 is reformulated into an empirical form suitable for adversarial training, resulting in the generation of the protected output. The overall STM pipeline is shown in Figure 1.

**Frequency-domain Perturbation**  We perform adversarial protection by introducing perturbations in the frequency domain. The raw speech waveform is converted into a spectrogram using STFT. In particular, we employ a Hann window with a $50\%$ overlap between consecutive frames. This configuration satisfies the COLA condition, ensuring that the sum of overlapping windows remains constant over time. The COLA property guarantees that the signal can be reliably reconstructed through iSTFT, which is critical for our framework, as perturbed spectrograms must be transformed back into time-domain waveforms without introducing additional artifacts.

We denote the STFT and iSTFT operations by $STFT : \mathbb{R}^* \to \mathbb{R}^{*\times*}$ and $iSTFT : \mathbb{R}^{*\times*} \to \mathbb{R}^*$, respectively, and write the spectrogram of $\boldsymbol{x} \in \mathbb{X}_u$ as $\boldsymbol{Y}_{\text{spec}} = STFT(\boldsymbol{x})$. For notational simplicity, we retain only the magnitude component and omit the phase, which remains unchanged throughout the algorithm. We denote the corresponding frequency-domain perturbation as $\boldsymbol{\Delta}_{\text{spec}}$, which shares the same shape as $\boldsymbol{Y}_{\text{spec}}$.

**Approximate Perceptibility Constraint**  To directly control the audibility of added noise, it is essential to quantitatively approximate the distortion perceived by humans between the raw speech audio and its protected version. In STM, we employ a psychoacoustic model, denoted as $Psyac(\boldsymbol{Y}_{\text{spec}}) : \mathbb{R}^{*\times*} \to \mathbb{R}^{*\times*}$, to estimate the masking threshold. This threshold, combined with predefined tolerance margins, provides an approximate representation of the human perceptual constraint in Equation 1.

Specifically, let $\boldsymbol{Y}_{\text{mask}} = Psyac(\boldsymbol{Y}_{\text{spec}})$ denote the masking threshold of the spectrogram $\boldsymbol{Y}_{\text{spec}}$. We then identify audible and inaudible tones in the original utterance as follows:

$$
\begin{aligned}
\mathbf{1}_{\text{audible}} &= \mathbf{1}_{\geq \boldsymbol{Y}_{\text{mask}}}(\boldsymbol{Y}_{\text{spec}}), \\
\mathbf{1}_{\text{inaudible}} &= \mathbf{1}_{< \boldsymbol{Y}_{\text{mask}}}(\boldsymbol{Y}_{\text{spec}}),
\end{aligned}
$$

where $\mathbf{1}_{\text{condition}}(\cdot)$ is an element-wise indicator function. We then impose the approximate perceptibility constraint on the protected audio as:

$$
\begin{aligned}
\boldsymbol{Y}_{\text{spec}}^{\text{adv}} &\leq \boldsymbol{Y}_{\text{spec}} + \mathbf{1}_{\text{audible}} \odot h_{\text{audible}}^{\text{head}} + \mathbf{1}_{\text{inaudible}} \odot h_{\text{inaudible}}^{\text{head}} := \boldsymbol{H}^{\text{head}}, \\
\boldsymbol{Y}_{\text{spec}}^{\text{adv}} &\geq \boldsymbol{Y}_{\text{spec}} + \mathbf{1}_{\text{audible}} \odot h_{\text{audible}}^{\text{floor}} + \mathbf{1}_{\text{inaudible}} \odot h_{\text{inaudible}}^{\text{floor}} := \boldsymbol{H}^{\text{floor}},
\end{aligned}
\tag{2}
$$

where $\boldsymbol{Y}_{\text{spec}}^{\text{adv}} = \boldsymbol{Y}_{\text{spec}} + \boldsymbol{\Delta}_{\text{spec}}$ denotes the spectrogram of the protected utterance, $\odot$ represents element-wise multiplication, and $h_{\{\text{audible},\text{inaudible}\}}^{\{\text{head},\text{floor}\}}$ correspond to the headroom and floorroom tolerances for audible and inaudible tones, respectively. These tolerance values are tunable parameters of STM that explicitly control the trade-off between audio quality and protection strength. For instance, by setting

$$
h_{\text{audible}}^{\text{floor}} = 0, \quad h_{\text{audible}}^{\text{head}} = 0, \quad h_{\text{inaudible}}^{\text{floor}} = -\infty, \quad h_{\text{inaudible}}^{\text{head}} = \boldsymbol{Y}_{\text{mask}} - \boldsymbol{Y}_{\text{spec}},
$$

STM enforces *perfectly stealthy* protection, where audible tones remain unchanged and inaudible tones are constrained strictly below the masking threshold.

Alternatively, by allowing controlled distortion through:

$$h_{\text{audible}}^{\text{floor}} = -c, \quad h_{\text{audible}}^{\text{head}} = c, \quad h_{\text{inaudible}}^{\text{floor}} = -\infty, \quad h_{\text{inaudible}}^{\text{head}} = \boldsymbol{Y}_{\text{mask}} - \boldsymbol{Y}_{\text{spec}} + c,$$

STM relaxes the constraint on audible tones by a margin $c > 0$, thereby increasing the protection strength at the cost of a small, controlled reduction in perceptual quality.

**Hard Clamping**  An effective approach used in this study to enforce the perceptibility constraint in Equation 2 is to hard clamp excessive values of $\boldsymbol{Y}_{\text{spec}}^{\text{adv}}$ (and thus of $\boldsymbol{\Delta}_{\text{spec}}$) back within the headroom and floorroom tolerances at the end of each training iteration:

$$Clamp(\boldsymbol{Y}_{\text{spec}}^{\text{adv}}) = \min\big(\max(\boldsymbol{Y}_{\text{spec}}^{\text{adv}}, \boldsymbol{H}^{\text{floor}}), \boldsymbol{H}^{\text{head}}\big),$$

where $\min$ and $\max$ are applied element-wise, returning the smaller or larger value of their two operands, respectively.

**Targeted Embedding Shift**  The ultimate objective of our problem is to reduce the confidence score produced by the SV model. However, directly optimizing the objective function in Equation 1 is inefficient and, in the case of autoregressive voice cloning models, often infeasible. Instead, we optimize the perturbation $\boldsymbol{\Delta}_{\text{spec}}$ to alter the output of the speaker encoder, eventually lower the SV confidence score.

In prior work on white-box adversarial protection methods, two primary objectives have been employed to deviate the output of the speaker encoder: *threshold-based* and *target-based*. The threshold-based approach trains the perturbation to push the speaker encoder's output a sufficient distance away from the original speaker embedding under a chosen distance metric, with the protection rate controlled by a predefined threshold. In contrast, the target-based approach forces the perturbed embedding to resemble that of another speaker selected from an embedding bank. In STM, we adopt the target-based approach, as guiding perturbations toward a valid subspace of real speech embeddings consistently yields higher audio quality in the protected outputs. Additional analysis of this effect is presented in Section 4.6.

**Multi-Step Total Variation Loss**  To further enhance the temporal smoothness of perturbations and thereby improve audio quality, we incorporate a *multi-step* one dimensional total variation (TV) loss into the optimization objective. Rather than penalizing only differences between adjacent samples, this formulation considers multiple step sizes to capture variations across different temporal scales. Formally, the multi-step TV loss is defined as:

$$Loss_{\text{MTV}} = \sum_{k \in \mathcal{K}} \frac{1}{T} \sum_{t=1}^{T} \big| \boldsymbol{\Delta}_{\text{spec}}[:, t+k] - \boldsymbol{\Delta}_{\text{spec}}[:, t] \big|,$$

where $\mathcal{K}$ is the set of step sizes, and $T$ denotes the length of $\boldsymbol{\Delta}_{\text{spec}}$ along the time axis. By enforcing smoothness across both short- and long-range time intervals, this regularization reduces rapid oscillations and stabilizes the perturbation structure.

The complete STM protection procedure is summarized in Algorithm 1.

# 4 EXPERIMENTS

## 4.1 DATASET

We evaluate our proposed protection method and baseline methods on the CSTR VCTK Corpus (Yamagishi et al., 2019). For preprocessing, we first discard utterances without transcripts, and then randomly select 100 speakers, with 5 utterances randomly chosen from each, resulting in a total of 500 utterances. From the remaining utterances, we select the longest one from each speaker to construct the embedding bank for target-based adversarial training. All audio files are downsampled to 16 kHz prior to training or inference with any model.

## 4.2 TARGET MODELS AND BASELINES

To best resemble a powerful attacker trying to clone the voice identity of the victim, we select four popular open-source DNN-based voice synthesis models with state-of-the-art zero-shot or few-shot voice cloning capabilities. The selected models are: GPT-SoVITS (GSV, RVC-Boss (2025)), FreeVC (Li et al., 2023b), QuickVC (Guo et al., 2023) and TriAAN-VC (Park et al., 2023).

---

**Algorithm 1:** Silence-the-Mimic (STM)

---

**Require:** speaker encoder $enc_{\text{spk}}$; STFT $STFT$; inverse STFT $iSTFT$; psychoacoustic
        model $Psyac$; clamp function $Clamp$; params $h_{\{\text{audible,inaudible}\}}^{\{\text{head,floor}\}}$; loss $Loss$

**Input:** victim utterance $\boldsymbol{x}$

**Output:** protected utterance $\boldsymbol{x}_{\text{adv}}$

1: $\boldsymbol{Y}_{\text{spec}} \leftarrow STFT(\boldsymbol{x})$
2: $\boldsymbol{Y}_{\text{mask}} \leftarrow Psyac(\boldsymbol{Y}_{\text{spec}})$
3: $emb_{\text{spk}}^{\text{orig}} \leftarrow enc_{\text{spk}}(\boldsymbol{x})$
4: Choose $emb_{\text{spk}}^{\text{trg}}$ from bank using $emb_{\text{spk}}^{\text{orig}}$
5: Compute $h^{\text{head}}, h^{\text{floor}}$ via Equation 2
6: Initialize $\boldsymbol{\Delta}_{\text{spec}}$; initialize optimizer $\mathcal{O}$
7: **for** $t \leftarrow 1$ **to** $T_{\text{iter}}$ **do**
8:      $\boldsymbol{Y}_{\text{spec}}^{\text{adv}} \leftarrow \boldsymbol{Y}_{\text{spec}} + \boldsymbol{\Delta}_{\text{spec}}$
9:      $\boldsymbol{x}_{\text{adv}} \leftarrow iSTFT(\boldsymbol{Y}_{\text{spec}}^{\text{adv}})$
10:     $emb \leftarrow enc_{\text{spk}}(\boldsymbol{x}_{\text{adv}}), \;\; loss \leftarrow Loss(emb, emb_{\text{spk}}^{\text{trg}}), \;\; \nabla_{\boldsymbol{\Delta}_{\text{spec}}} \leftarrow \partial loss / \partial \boldsymbol{\Delta}_{\text{spec}}$
11:     $\boldsymbol{\Delta}_{\text{spec}} \leftarrow \mathcal{O}\text{-update}(\boldsymbol{\Delta}_{\text{spec}}, \nabla_{\boldsymbol{\Delta}_{\text{spec}}})$
12:     $\boldsymbol{\Delta}_{\text{spec}} \leftarrow Clamp(\boldsymbol{\Delta}_{\text{spec}}; \boldsymbol{Y}_{\text{spec}}, \boldsymbol{Y}_{\text{mask}}, h^{\text{head}}, h^{\text{floor}})$
13: **return** $x_{\text{adv}}$

---

We evaluate the defense performance of our proposed method on the models described above, comparing it with two baselines: AntiFake (Yu et al., 2023) and VoiceGuard (Li et al., 2023a). Both methods are widely recognized and continue to serve as baselines in recent studies (Gao et al., 2025; Fan et al., 2025), underscoring their status as current leading approaches. They operate under a white-box setting and employ adversarial training to generate perturbations in the time domain, combined with loss based audio quality control. The key distinction lies in their strategies for quality control: AntiFake integrates the loss terms directly into the perturbation process, whereas Voice-Guard performs audio refinement as a separate stage following adversarial training, with the optimal weight determined via binary search.

### 4.3 Evaluation Metrics

We evaluate the protection effectiveness of STM and the baseline methods using three widely adopted SV models: TitaNet (Koluguri et al., 2022), ECAPA-TDNN (Desplanques et al., 2020), and Pyannote.audio (Plaquet & Bredin, 2023; Bredin, 2023). For TitaNet and ECAPA-TDNN, we employ the implementations provided in NVIDIA's `NeMo` library, while for Pyannote.audio, we use the official implementation from its official repository.

Following prior work, we evaluate the audio quality of protected utterances using the Mean Opinion Score (MOS) and the Noisiness metric. MOS provides an overall measure of perceived speech quality, while the Noisiness metric quantifies degradation due to additive noise, capturing the extent to which an audio sample is perceived as "not noisy" (Wältermann, 2013; Mittag et al., 2021). In addition, we incorporate the Perceptual Evaluation of Speech Quality (PESQ) (Rix et al., 2001) to assess voice naturalness, and the Short-Time Objective Intelligibility (STOI) (Taal et al., 2010) to evaluate speech intelligibility. Together, these metrics provide a comprehensive evaluation by capturing complementary aspects of perceptual quality, ranging from naturalness to intelligibility and noise robustness. MOS and Noisiness are obtained using NISQA (Mittag et al., 2021), an open-source DNN-based model for multidimensional speech quality prediction. For PESQ and STOI, we adopt the `pesq`[1] and `pystoi`[2] libraries, respectively.

---

[1] https://github.com/ludlows/PESQ
[2] https://github.com/mpariente/pystoi

## 4.4 TRAINING DETAILS

For our proposed method, we configure the STFT with 512 FFT size and a hop size of 256 to satisfy the COLA condition. The resulting spectrograms are subsequently converted into the log scale. The psychoacoustic model is adopted from Schuller (2020). We extract the speaker encoder components from GSV, FreeVC, QuickVC, and TriAAN-VC, and employ them as $enc_{\mathrm{spk}}$ in Algorithm 1. The target speaker embedding, $emb_{\mathrm{spk}}^{\mathrm{trg}}$, is selected as the embedding corresponding to the fifth-largest distance from the original speaker embedding, $emb_{\mathrm{spk}}^{\mathrm{orig}}$, in the entire speaker bank. Perceptibility control parameters are set as follows:

$$h_{\mathrm{audible}}^{\mathrm{floor}} = -0.15, \quad h_{\mathrm{audible}}^{\mathrm{head}} = 0.15, \quad h_{\mathrm{inaudible}}^{\mathrm{floor}} = -\infty, \quad h_{\mathrm{inaudible}}^{\mathrm{head}} = \boldsymbol{Y}_{\mathrm{mask}} - \boldsymbol{Y}_{\mathrm{spec}}.$$

We use the Adam optimizer with a learning rate of 1 and parameters $\beta_1 = 0.9$ and $\beta_2 = 0.999$. $T_{\mathrm{iter}}$ is set to 80. For loss function, we use a combination of L1 loss and cosine similarity loss as shown in Equation 3:

$$Loss(emb, emb_{\mathrm{spk}}^{\mathrm{trg}}) = ||emb - emb_{\mathrm{spk}}^{\mathrm{trg}}||_1 + \big(1 - CosineSimilarity(emb, emb_{\mathrm{spk}}^{\mathrm{trg}})\big). \tag{3}$$

TV loss is incorporated into Equation 3 for defense against FreeVC and QuickVC.

For the baselines, we evaluate AntiFake under two configurations: 500 and 1000 iterations, with the latter corresponding to its original setting. We also manually adjust the weight ratio between the speaker loss and the quality control loss to balance audio fidelity and defense robustness. Additional evaluations of AntiFake with different quality control loss weights are provided in Section 4.6. For VoiceGuard, we adopt the configuration from the original paper, consisting of 3000 adversarial training steps followed by 1500 refinement steps.

For the three SV models, the thresholds of normalized cosine similarity are set to 0.7318, 0.6466, and 0.7507 for ECAPA-TDNN, Pyannote, and TitaNet, respectively. These thresholds are determined by their Equal Error Rates (EER) on the VCTK dataset.

All methods are evaluated on a single NVIDIA A100 80GB PCIe GPU. Training time is averaged over the entire sampled dataset, accounting only for perturbation iterations and excluding data I/O.

## 4.5 MAIN RESULTS

Table 1 reports the comparative SV rejection rates and audio quality metrics. TitaNet, E-TDNN, and Pyannote denote the rejection rates obtained from the TitaNet, ECAPA-TDNN, and Pyannote.audio SV systems, respectively. F-VC, Q-VC, and T-VC denote the target models FreeVC, QuickVC, and TriAAN-VC, respectively. STM denotes our proposed method, AF-500 and AF-1000 correspond to AntiFake with 500 and 1000 iterations, and VG represents VoiceGuard. Confidence intervals are reported at the 95% level. The symbol ↑ indicates that higher values are preferable. The best results are shown in **bold**, and the second best results are underlined.

Across all VC and TTS models, STM consistently demonstrates state-of-the-art protection performance. It achieves the highest or second-highest rejection rates on nearly all SV systems, frequently surpassing VoiceGuard and AntiFake by large margins. With protection performance equal to or exceeding that of the baselines, STM preserves superior perceptual quality in the protected audio. It achieves above-average scores in MOS and PESQ, while its STOI and Noisiness results are consistently higher than those of the baselines, indicating more effective perceptual masking of perturbations.

Most notably, STM delivers an order-of-magnitude improvement in inference speed compared to existing baselines. As shown in Table 2, it reduces inference time from tens or even hundreds of seconds per iteration to only a few seconds, representing at least a $20\times$ speedup across all settings.

## 4.6 FURTHER EXPERIMENTS

**Effect of Quality Control Loss Weight in AntiFake** In Table 1, we report the protection performance of AntiFake under varying loss weight ratios between the speaker encoder loss and the quality control loss. To enable a more comprehensive performance comparison, we further evaluate AntiFake-500 across a broader range of loss weight ratios, with the corresponding results presented

Table 1: Comparison of SV rejection rates and audio quality metrics between the proposed method and the baseline methods.

| Model | Method | TitaNet ↑ | E-TDNN ↑ | Pyannote ↑ | MOS ↑ | Noisiness ↑ | PESQ ↑ | STOI ↑ |
|---|---|---|---|---|---|---|---|---|
| GSV | STM | **96.00%** | **95.40%** | 68.80% | **3.26 ± 0.06** | **3.65 ± 0.04** | **2.54 ± 0.03** | **0.91 ± 0.00** |
| | AF-500 | 83.20% | 86.40% | 66.80% | 2.98 ± 0.04 | 1.89 ± 0.02 | 1.86 ± 0.03 | 0.79 ± 0.01 |
| | AF-1000 | 80.00% | 72.60% | 48.40% | 3.25 ± 0.06 | 2.03 ± 0.04 | 2.00 ± 0.04 | 0.82 ± 0.01 |
| | VG | 94.00% | 95.20% | **81.40%** | 2.82 ± 0.05 | 1.78 ± 0.02 | 1.72 ± 0.02 | 0.78 ± 0.01 |
| F-VC | STM | **98.40%** | **98.80%** | 94.40% | **3.80 ± 0.07** | **3.82 ± 0.05** | **2.86 ± 0.03** | **0.94 ± 0.00** |
| | AF-500 | 76.80% | 77.60% | 70.40% | 3.22 ± 0.05 | 2.12 ± 0.03 | 1.95 ± 0.03 | 0.81 ± 0.01 |
| | AF-1000 | 76.20% | 77.20% | 69.40% | 3.26 ± 0.11 | 2.30 ± 0.09 | 2.04 ± 0.06 | 0.80 ± 0.03 |
| | VG | 83.20% | 84.40% | 76.00% | 2.99 ± 0.04 | 1.93 ± 0.02 | 1.79 ± 0.02 | 0.78 ± 0.01 |
| Q-VC | STM | **99.60%** | **99.60%** | 98.80% | 3.64 ± 0.08 | **3.67 ± 0.06** | **2.59 ± 0.03** | **0.95 ± 0.00** |
| | AF-500 | 57.20% | 56.80% | 51.60% | 3.22 ± 0.04 | 2.15 ± 0.01 | 2.20 ± 0.03 | 0.83 ± 0.01 |
| | AF-1000 | 51.00% | 51.20% | 41.20% | 3.55 ± 0.04 | 2.45 ± 0.02 | 2.58 ± 0.03 | 0.86 ± 0.01 |
| | VG | 97.80% | 97.80% | 95.40% | 2.90 ± 0.04 | 1.74 ± 0.01 | 1.73 ± 0.02 | 0.77 ± 0.01 |
| T-VC | STM | 67.80% | 65.60% | 43.40% | 2.86 ± 0.06 | **3.54 ± 0.03** | **2.36 ± 0.03** | **0.89 ± 0.01** |
| | AF-500 | 67.40% | 68.00% | **46.00%** | 2.86 ± 0.04 | 1.81 ± 0.01 | 1.87 ± 0.03 | 0.79 ± 0.01 |
| | AF-1000 | 67.40% | 67.20% | **46.00%** | 2.98 ± 0.03 | 1.91 ± 0.01 | 2.03 ± 0.03 | 0.80 ± 0.01 |
| | VG | **69.20%** | **67.60%** | 42.80% | 2.89 ± 0.04 | 1.92 ± 0.01 | 1.97 ± 0.03 | 0.83 ± 0.01 |

Table 2: Inference speed across models and methods.

| Model | Speed ↓ (s/it) | | | |
|---|---|---|---|---|
| | STM | AF-500 | AF-1000 | VG |
| GSV | **1.02** | 18.35 | 38.34 | 30.87 |
| F-VC | **2.01** | 20.10 | 43.38 | 56.80 |
| Q-VC | **1.76** | 21.02 | 42.55 | 79.72 |
| T-VC | **3.96** | 27.85 | 57.53 | 156.27 |

in Table 3. Column *QC Weight* specifies the applied adjustment to the quality control loss. The notation ×c indicates that the loss weight is scaled by the factor $c$, while a value of 1 denotes direct substitution of the target model's speaker encoder in AntiFake. For FreeVC, a QC weight of ×0.5 is equivalent to drop-in and is therefore not reported.

Table 3: Effect of varying the quality control loss weight in AntiFake.

| Model | QC weight | TitaNet ↑ | E-TDNN ↑ | Pyannote ↑ | MOS ↑ | Noisiness ↑ | PESQ ↑ | STOI ↑ |
|---|---|---|---|---|---|---|---|---|
| GSV | ×2 | 68.60% | 73.40% | 47.80% | 2.95 ± 0.04 | 1.99 ± 0.02 | 2.07 ± 0.03 | 0.82 ± 0.01 |
| | ×0.5 | 96.40% | 96.60% | 89.60% | 2.85 ± 0.04 | 1.69 ± 0.02 | 1.59 ± 0.02 | 0.75 ± 0.01 |
| | 1 (drop-in) | 100.00% | 100.00% | 99.60% | 2.02 ± 0.03 | 1.46 ± 0.01 | 1.09 ± 0.00 | 0.60 ± 0.01 |
| F-VC | ×5 | 2.76% | 3.31% | 5.52% | 3.80 ± 0.05 | 2.86 ± 0.06 | 2.99 ± 0.05 | 0.89 ± 0.01 |
| | ×2 | 54.60% | 55.20% | 47.00% | 3.32 ± 0.04 | 2.41 ± 0.02 | 2.30 ± 0.04 | 0.84 ± 0.01 |
| | ×0.5 | — | — | — | — | — | — | — |
| | 1 (drop-in) | 95.80% | 95.00% | 90.60% | 1.57 ± 0.03 | 1.37 ± 0.01 | 1.22 ± 0.01 | 0.69 ± 0.01 |
| Q-VC | ×2 | 53.00% | 54.00% | 46.40% | 3.22 ± 0.04 | 2.17 ± 0.01 | 2.23 ± 0.03 | 0.83 ± 0.01 |
| | ×0.5 | 63.00% | 62.40% | 55.20% | 3.26 ± 0.04 | 2.10 ± 0.01 | 2.15 ± 0.03 | 0.83 ± 0.01 |
| | 1 (drop-in) | 97.00% | 97.20% | 93.20% | 2.02 ± 0.03 | 1.46 ± 0.01 | 1.09 ± 0.00 | 0.60 ± 0.01 |
| T-VC | ×2 | 99.60% | 99.80% | 98.00% | 1.49 ± 0.02 | 1.42 ± 0.01 | 1.13 ± 0.01 | 0.66 ± 0.01 |
| | ×0.5 | 90.20% | 90.60% | 76.60% | 2.60 ± 0.04 | 1.72 ± 0.01 | 1.70 ± 0.02 | 0.76 ± 0.01 |
| | 1 (drop-in) | 100.00% | 99.80% | 100.00% | 1.12 ± 0.01 | 1.47 ± 0.01 | 1.07 ± 0.00 | 0.58 ± 0.01 |

As shown in the table, for GSV, FreeVC, and QuickVC, when the protection rate approaches STM, their audio quality degrades substantially compared to our proposed method. Moreover, as the weight of the quality control loss increases, the protection rate decreases even further relative to

STM, underscoring the effectiveness of our approach. For TriAAN-VC, however, adjusting the quality control weight has inverse effect. This may be attributed to the distinct nature of the TriAAN-VC speaker encoder, which produces variable-length speaker embeddings. Such representations make loss-based quality control less effective, thereby highlighting STM's superiority in enforcing audio distortion tolerance.

**Effect of Targeted Embedding Shift** To examine the effect of targeted embedding shift on audio quality and model performance, we modify the loss function in Equation 3 by replacing $emb_{\mathrm{spk}}^{\mathrm{trg}}$ with $emb_{\mathrm{spk}}^{\mathrm{orig}}$ and reversing the sign of each term. Table 4 reports the STM evaluation results with the modified loss function, showing a substantial degradation in audio quality for GSV, FreeVC, and QuickVC compared with the targeted embedding shift. This demonstrates that under our perceptibility control method, guiding perturbations toward a valid subspace of speaker embeddings improves the quality of the generated audio.

Table 4: Evaluation of STM without directed embedding shift.

| Model | TitaNet ↑ | E-TDNN ↑ | Pyannote ↑ | MOS ↑ | Noisiness ↑ | PESQ ↑ | STOI ↑ |
|---|---|---|---|---|---|---|---|
| GSV | 97.40% | 97.80% | 67.20% | $2.68 \pm 0.06$ | $3.50 \pm 0.04$ | $2.22 \pm 0.03$ | $0.88 \pm 0.00$ |
| F-VC | 99.80% | 99.20% | 97.40% | $3.38 \pm 0.06$ | $2.72 \pm 0.06$ | $2.22 \pm 0.02$ | $0.91 \pm 0.00$ |
| Q-VC | 99.40% | 99.00% | 97.40% | $3.42 \pm 0.06$ | $2.88 \pm 0.07$ | $2.22 \pm 0.02$ | $0.93 \pm 0.00$ |
| T-VC | 51.60% | 51.60% | 20.40% | $2.84 \pm 0.06$ | $3.54 \pm 0.04$ | $2.52 \pm 0.03$ | $0.86 \pm 0.01$ |

**Effect of Multi-Step TV Loss** We remove the multi-step TV losses from the objective function, and the corresponding evaluation results are reported in Table 5. As shown, without TV loss the protected utterances contain abruptly varying noise. These ineffective perturbations produce noticeable artifacts in the audio, thereby reducing the practicality of the protection.

Table 5: Evaluation of STM without TV loss.

| Model | TitaNet ↑ | E-TDNN ↑ | Pyannote ↑ | MOS ↑ | Noisiness ↑ | PESQ ↑ | STOI ↑ |
|---|---|---|---|---|---|---|---|
| F-VC | 99.40% | 99.60% | 95.80% | $2.74 \pm 0.07$ | $3.04 \pm 0.07$ | $2.34 \pm 0.02$ | $0.92 \pm 0.00$ |
| Q-VC | 99.80% | 99.60% | 98.20% | $2.83 \pm 0.07$ | $3.00 \pm 0.07$ | $2.24 \pm 0.02$ | $0.93 \pm 0.00$ |

## 5 CONCLUSION

In this paper, we introduced a fast and effective method for protecting speech from voice cloning attacks. By perturbing audio in the frequency domain under the guidance of a psychoacoustic model, our approach enforces strict perceptual limits while greatly accelerating adversarial perturbation generation. Experiments on state-of-the-art voice cloning models show that our method delivers competitive protection with higher audio quality and substantially reduced processing time.

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

## A    LLM USAGE

In accordance with the ICLR 2026 policy on large language model (LLM) usage, we disclose that a general-purpose LLM was employed exclusively for language polishing of the manuscript draft. The LLM's role was limited to improving grammar, clarity, and style.

All research ideas, conceptual development, methodology design, experiments, analyses, and conclusions were independently conducted by the authors without LLM involvement. The use of the LLM did not extend to ideation, experimental design, data analysis, or result interpretation.

