# OpenReview forum: "Silence-the-Mimic: Accelerating Imperceptible Perturbations Against Voice Cloning"
_ICLR.cc/2026/Conference — Submitted to ICLR 2026_

### Official Review · Reviewer_8Y2j · 2025-10-26

**Soundness:** 3
**Presentation:** 3
**Contribution:** 3
**Rating:** 4
**Confidence:** 4

**Summary:**

The paper proposes Silence-the-Mimic (STM), a protection method that constructs adversarial perturbations directly in the STFT (frequency) domain under a psychoacoustic masking constraint. STM clamps spectrogram perturbations to remain below masking thresholds (with tunable head/floor margins), applies multi-step total variation regularization, and trains perturbations to shift speaker-encoder embeddings toward a target embedding to cause speaker-verification rejection. Experiments on several state-of-the-art VC/TTS models (GPT-SoVITS, FreeVC, QuickVC, TriAAN-VC) and three SV backends show high rejection rates, good perceptual scores, and substantial speedups versus time-domain baselines (reported inference/training speeds)

**Strengths:**

Practical and fast frequency-domain design. The idea to clamp spectrogram updates to psychoacoustic masking thresholds is simple and computationally efficient; the paper reports large speedups (order-of-magnitude) compared to prior iterative time-domain adversarial approaches.

Comprehensive empirical coverage within the chosen scope. STM is evaluated across multiple VC/TTS victim models and three common SV detectors and compares to established baselines (AntiFake, VoiceGuard), with many perceptual metrics (MOS, PESQ, STOI, Noisiness) shown.

Careful perceptual control (psychoacoustic clamping + TV loss). The hard-clamp mechanism and multi-step TV loss are reasonable engineering choices to keep perturbations smooth and (claimed) imperceptible.

**Weaknesses:**

A. Visibility of perturbation in the frequency domain is  not analyzed or visualized
STM directly optimizes spectrogram magnitudes (∆spec) and clamps them against a mask. Optimizing in magnitude space tends to produce structured spectral footprints (lines/bands) that are visually obvious in spectrograms and can be audible after iSTFT phase interactions, even if masked numerically. The paper does not show spectrogram visualizations of ∆spec, nor any psychophysical listening validation beyond MOS/PESQ. If the perturbation is visually or algorithmically detectable in frequency space, an adversary or downstream defense could filter or flag such protected audio, negating the protection. Also, a visible frequency footprint undermines the “imperceptible” claim even if SNR/MOS numbers look acceptable.

The robustness of the proposed protection is not evaluated. In real-world scenarios, protected audio may undergo transformations such as resampling, compression, or noise addition, and attackers may retrain their models using both protected and unprotected data. Without demonstrating that the proposed method remains effective under such conditions, the “robust protection” claim is unconvincing and cannot be compared fairly with prior methods that explicitly address black-box and distortion robustness.

The efficiency claim in the paper is incomplete. While the authors report faster generation time, the comparison lacks details about the total training time and full end-to-end latency. I remember the AntiFake is black-box protection, in this case, the AntiFake needs to optimize with different group of models to enhance the transferbility. This is a more challenging scenario compared to the author's paper, therefore, I suspect the comparison in efficiency is unfari. A fair evaluation should include the total time required to produce a complete protected sample and compare it against existing methods using the same computational settings.

The paper repeatedly claims its effectiveness on TTS scenarios but only evaluates one TTS model. It does not explore training-based TTS or VC attacks, where an adversary could fine-tune or retrain a model using both protected and clean samples. Without such evaluation, it is unclear whether the proposed defense remains valid in realistic training-based cloning cases.

The paper lacks released code or demonstration. Since the evaluation involves perceptual quality and inaudibility claims, reproducibility and transparency are important. The absence of public code or demo samples makes it difficult to verify the reported results and to compare with prior work.

**Questions:**

Could the authors visualize and analyze the perturbation in the frequency domain to verify that it is indeed imperceptible?

How does the proposed method perform under black-box settings, where the attacker uses different encoders or TTS models?

Has the robustness been tested under common distortions such as MP3 compression, resampling, or additive noise?

What is the total wall-clock time to generate one protected audio sample compared to AntiFake or VSMask?

Would the protection remain effective if the attacker collects both protected and unprotected samples to train a new TTS or VC model?

Are there plans to release code or provide a demo for verification of the reported perceptual results?

---

### Official Review · Reviewer_GC2j · 2025-10-29

**Soundness:** 2
**Presentation:** 1
**Contribution:** 2
**Rating:** 4
**Confidence:** 3

**Summary:**

The paper proposes a protection algorithm, Silence-the-Mimic (STM), aimed to address the DeepFake audio problem. The authors introduce an imperceptible and fast protection method, by injecting perturbation in the frequency domain under hard constraints of a psychoacoustic model.

**Strengths:**

- The paper addresses a meaningful real-world problem with clear applications
- Problem formulation is clear and concise
- The paper demonstrates improvements over the baselines, achieving better rejection rates and audio quality, while maintaining faster inference

**Weaknesses:**

- More commentary on the motivation behind each design choice would help to clarify the advantages of the proposed method, compared to the VoiceGuard baseline for example, which similarly uses a psychoacoustic model in the time domain to mitigate issues associated with frequency domain perturbations.
- The method replaces an iterative loss-weighting search with manually fixed tolerance margins, which still require tuning for each model or task, potentially limiting generalizability.
- The paper would benefit from a discussion of the practical limitations of the proposed protection algorithm. It would be helpful if the authors could identify specific cases or model types where the suggested protection may be less effective.

**Questions:**

1. Could you please provide an ablation of the headroom and floorroom tolerances and how they affect protection performance?
2. In Table 5, why not all models are shown for the TV loss ablation? The motivation for using this term is clear and I would expect unnatural audio artifacts for all models.
3. The method operates only on the spectrogram magnitude, leaving the phase unchanged. Would it potentially leave a vulnerability space for future attacks that can estimate the phase?
4. Could the authors discuss the limitations of their work and how they plan to address them in future work?

---

### Official Review · Reviewer_4K1U · 2025-10-30

**Soundness:** 2
**Presentation:** 3
**Contribution:** 2
**Rating:** 2
**Confidence:** 4

**Summary:**

This paper proposes Silence-the-Mimic, a frequency-domain defense method designed to prevent unauthorized voice cloning by introducing imperceptible perturbations under psychoacoustic masking constraints. Unlike prior time-domain adversarial defenses, STM applies perturbations directly in the spectrogram space, enforces perceptual limits through a psychoacoustic model, and claims an order-of-magnitude speedup while maintaining comparable protection performance. Experiments on several voice cloning models demonstrate similar or slightly improved speaker verification rejection rates with higher MOS and PESQ scores.

**Strengths:**

- The psychoacoustic masking–based constraint is conceptually sound and aligns with established principles in human perception.

- The paper is logically clear and easy to follow.

**Weaknesses:**

- One of the core claimed contributions is the computational speedup. However, the practical importance of this improvement is unclear. Voice protection is typically executed offline by the defender before publishing their audio data, not under real-time constraints. Thus, shaving off tens of seconds during preprocessing offers negligible benefit in real-world usage. Unlike attacks, which require rapid inference to evade detection or interact with APIs, defenders can afford slower but more thorough optimization. The paper should justify why speed is an essential property for protection methods and whether it truly enhances usability or deployment scalability.

- The paper attributes the acceleration to optimization in the frequency domain and the removal of iterative quality-control losses. However, this explanation is not technically convincing. The proposed pipeline still involves STFT/iSTFT conversions—operations that are computationally nontrivial—and iterative optimization over ∆spec. The paper does not provide ablation results isolating which components (e.g., Targeted Embedding Shift) account for the observed efficiency gains.

- While the idea of leveraging psychoacoustic masking to conceal adversarial perturbations is reasonable, it is not novel. Prior adversarial audio works have already employed similar perceptual masking principles (e.g., [r1]). STM essentially repackages these concepts in the voice-cloning defense context with minor procedural differences (target embedding shift). The contribution thus lies more in engineering refinement than in conceptual advancement.

- Although the paper emphasizes human perceptibility, its evaluation relies entirely on automatic metrics (MOS prediction, PESQ, STOI) rather than real human perception studies. Given that the main technical contribution involves perceptual imperceptibility, a small-scale human test would provide much stronger evidence of real-world usability. Without such validation, the results cannot conclusively demonstrate imperceptibility to human listeners.

[r1] Schönherr, Lea, et al. "Adversarial attacks against automatic speech recognition systems via psychoacoustic hiding." Network and Distributed Systems Security (NDSS) Symposium 2019.

**Questions:**

- Why is computational speedup important for a defense method that is typically applied offline?

- What specific components of the proposed pipeline contribute to the reported acceleration?

- How does the proposed psychoacoustic masking approach differ technically from prior adversarial audio attacks that also leverage perceptual masking?

- Would a user study or listening test provide more reliable evidence of imperceptibility in real-world settings?

---

### Official Review · Reviewer_PYSg · 2025-10-30

**Soundness:** 3
**Presentation:** 4
**Contribution:** 3
**Rating:** 6
**Confidence:** 3

**Summary:**

The rise of audio DeepFakes poses serious technical and ethical challenges for the field. The authors categorize existing work to defend against audio DeepFakes into two categories: post-hoc active/passive detection and noise-signal guided defenses. It is argued that these prior methods are difficult to control due to requiring multiple regularization terms, and require computationally expensive optimization algorithms. The authors propose Silence-the-Mimic (STM) which operates in the frequency domain instead of the time domain of the raw signal. Conceptually, the threat model assumes the victim publishes voice samples online, from which an attacker may exploit to generate deepfake audio with arbitrary content. A speaker verification model determines whether any audio, either original or deepfake, is authentic to the victim speaker. The STM pipeline first converts the sample audio to STFT frequency domain across 50% sliding windows in time to ensure COLA condition during the inverse operation. STM only modifies the magnitude component and leaves the phase alone. The total adversarial noise is controlled using a perceptibility mask generated by a psychoacoustic function for each STFT frame. The mask acts as an approximate representation of the optimization constraint in prior work, where the mask is parameterized jointly by audible and inaudible noise floor and noise ceiling.

The adversarial noise for each frame is tuned such that the speaker verification model misclassifies the generated audio as another speaker. In addition to the psychoacoustic model, STM also leverages a loss which penalizes large signal variations between frames at multiple time scales, ideally enforcing smoothness in the perturbation. STM is validated with audio samples from the CSTR VCTK dataset on four relatively recent VC models and compare against AntiFake and VoiceGuard as baselines. Three SV models are used for verifying attacked audio identity. In experiments, the authors check the voice quality with MOS, PESQ, STOI, and Noisiness metrics based on open-source implementations. The authors use an existing psychoacoustic model and present results comparing SV rejection rate, inference speed, and ablations on quality control.

**Strengths:**

- The submission targets a relevant problem in the protection and privacy of online audio data, highlighting the drawbacks of prior methods, and proposes reasonable solutions by way of new attack formulation and a new attack algorithm.
- The writing is polished and there are clear logical arguments motivating the core research problems, their solutions, and evaluation methodology. The tables and figures are rendered properly with no perceptible issues. Results are reported with a known confidence interval.
- The proposed STM attack surpasses the selected baselines and more importantly reports a significant decrease in inference time. The authors provide component ablations for TV loss and removing target speaker guidance, showing that these components are generally beneficial.

**Weaknesses:**

- The authors mainly focus on the white-box setting, but it would be insightful to check if the attack formulation improves attack transferability to unknown VC models and unknown speaker identities compared to the baselines. In that case, it isn't clear if faster inference time necessarily permits better transferability, since the adversarial noise may saturate the proposed mask region before the unknown decision boundary is crossed. The black-box setting is relevant here since the threat model assumes arbitrary voice samples submitted online, so it seems reasonable for the VC model to be unknown.
- There is an abundance of quality metrics in the reported results, but ultimately since VC models are used for fooling human users, the contribution would be improved with a human study to validate the assumptions related to leveraging the psychoacoustic model, such as the reported effectiveness of noise floor and noise ceiling.
- Some annotations in Algorithm 1 would be helpful to connect the description in text with each line of pseudocode.
- Expanding the results to include more datasets would offer insight for noise floor/ceiling settings.

**Questions:**

- For L334, can the authors clarify the choice of noise floors and noise ceilings? Are these settings adequate for all datasets, or only VCTK?

---

### Meta-Review · Area_Chair_5xuC · 2026-01-06

**Summary:**

The reviewers raised several substantial concerns that remain unaddressed due to the lack of author response. A central issue is the insufficient validation of the paper’s core claim of imperceptible protection: despite relying on psychoacoustic masking, the evaluation uses only automatic perceptual metrics and provides no human listening study, spectrogram analysis, or psychophysical validation. In addition, the novelty of the approach is unclear, as psychoacoustic masking has been widely explored in prior adversarial audio work, and the paper does not clearly distinguish its contributions beyond incremental engineering refinements. The evaluation scope is further limited by a predominantly white-box threat model, lack of robustness analysis under common audio transformations or training-based attacks, and insufficient coverage of TTS models. Finally, several design choices and hyperparameters are not adequately justified. Taken together, these unresolved issues prevent sufficient confidence in the paper’s claims, leading to a recommendation for rejection.

**Reviewer Concerns:**

The authors did not provide a response, so this part is not applicable.

**Reviewer Scores:**

The authors did not provide a response, so this part is not applicable.

---

### Decision · Program_Chairs · 2026-01-26

Reject